# Invariant Structure Learning for Better Generalization and Causal Explainability

**Yunhao Ge♦,★**; **Sercan Ö. Arık♦, Jinsung Yoon♦, Ao Xu★, Laurent Itti★,** and **Tomas Pfister♦**
*{yunhaoge, aoxu, itti}@usc.edu, {soarik, jinsungyoon, tpfister}@google.com*

♦ *Google Cloud AI, Sunnyvale, CA, USA*
★ *University of Southern California, Los Angeles, CA, USA*

*Reviewed on OpenReview:* `https://openreview.net/forum?id=A9yn7KTwsK`

## Abstract

Learning the causal structure behind data is invaluable for improving generalization and obtaining high-quality explanations. Towards this end, we propose a novel framework, Invariant Structure Learning (ISL), that is designed to improve causal structure discovery by utilizing generalization as an indication in the process. ISL splits the data into different environments, and learns a structure that is invariant to the target across different environments by imposing a consistency constraint. The proposed aggregation mechanism then selects the classifier based on a graph structure that reflects the causal mechanisms in the data more accurately compared to the structures learnt from individual environments. Furthermore, we extend ISL to a self-supervised learning setting, where accurate causal structure discovery does not rely on any labels. Self-supervised ISL utilizes proposals for invariant causality, by iteratively setting different nodes as targets. On synthetic and real-world datasets, we demonstrate that ISL accurately discovers the causal structure, outperforms alternative methods, and yields superior generalization for datasets with significant distribution shifts. We open-source our code at `https://github.com/AaronXu9/ISL.git`.[1]

## 1 Introduction

High capacity machine learning models such as deep neural networks (DNNs) have fueled transformational progress in numerous domains where the i.i.d. assumption is mostly valid (Zhang et al., 2021b) as they can be very effective in fitting to available training data. However, as a severe blind spot in conventional machine learning, the performance of such models can be much worse on the out-of-distribution (OOD) test data. This 'overfitting' phenomena can be attributed to over-parameterized models such as DNNs absorbing spurious correlations as shown in Fig. 1, from the training data and resulting in biases unrelated to the causal relationships that truly drive the input-output mapping for both training and test samples (Zhang et al., 2021a; Schott et al., 2021; Roelofs et al., 2019; D'Amour et al., 2020; Bartlett et al., 2021).

In most cases, the machine learning problems are underspecified, i.e. there are multiple distinct solutions that solve the problem by achieving equivalent held-out performance on i.i.d. data. Underspecification in practice can be an obstacle to reliable real-world deployment of high capacity machine learning models, as such models can exhibit unexpected behavior when the test data deviate from the training data (D'Amour et al., 2020; Arjovsky et al., 2019).

Various methods have been proposed towards reducing mitigating underspecification and overfitting: regularization approaches (Arpit et al., 2017; Ng, 2004; Tibshirani, 1996; Hoerl & Kennard, 1970) constrain

---

*Work done while at Google
[1]The implementation is available in `https://github.com/AaronXu9/ISL.git`

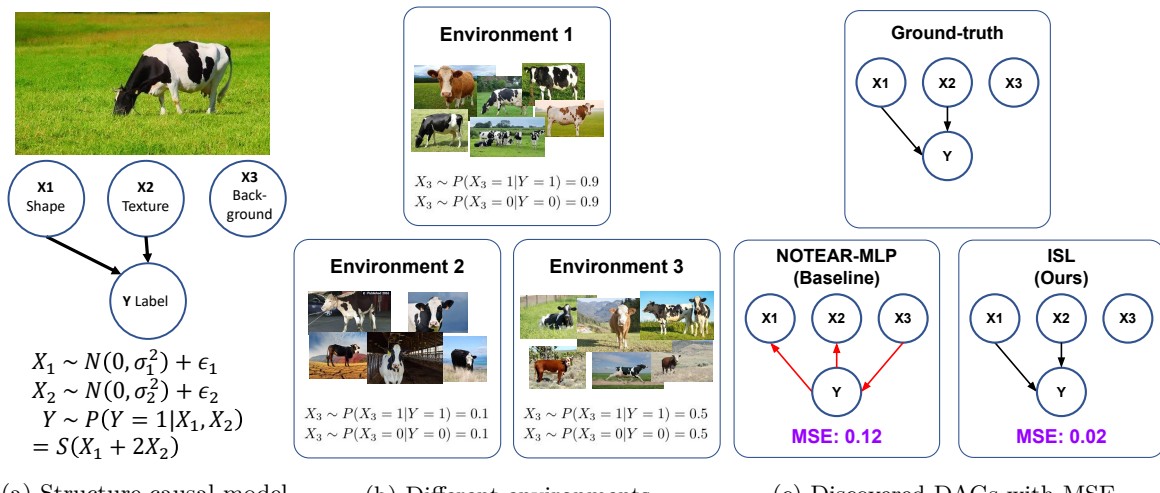

(a) Structure causal model          (b) Different environments          (c) Discovered DAGs with MSE.

Figure 1: A motivational example. (a) For the image label Y (1 means the label is "cow" and 0 otherwise), X1 and X2 represent the causal parents about the image details (here, shape and texture), and X3 (background type where 1 indicates the presence of grass and 0 otherwise) represents a factor that isn't causal to Y. S(·) is sigmoid function. In this example, texture (X2) is twice as causal to Y than shape (X1). (b) The relationship between Y and X3 vary across environments; since the conditional dependence is not consistent across environments X3 may not be treated as a major causal factor for Y. (c) We utilize the Mean Squared Error (MSE) as a metric to assess the prediction error for 'Y'. This is carried out by using the projected causal parent of 'Y' as features inputted into a two-layer neural network. Smaller MSE value implies that the causal parents variables used for prediction are more precise. Our proposed method ISL yields more accurate discovery of the underlying causal relation – here, it correctly identifies X1 and X2 but not X3 as the causal factors of Y, improving the explanation quality and prediction accuracy.

the flexibility of models; data augmentation methods (Yaeger et al., 1996; Krizhevsky et al., 2012) generate artificially-transformed samples invariant to labels; judicious DNN designs (Arik & Pfister, 2019; Oord et al., 2016; Jaderberg et al., 2015; Lim et al., 2019) introduce appropriate inductive biases for the data types of interest. Notably, the CASTLE (Kyono et al., 2020) approach introduces a novel regularization method that incorporates causal relationships into the regularization process.

These approaches have shown some progress in improving generalization, and in some cases they yield significant improvements in test accuracy, however, underlying systematic framework beneath them is missing – they do not tackle the fundamental challenge of discovering causal relationships that are consistent across the training and test data and basing the decision making on them. Thus, their improvements remain restricted to specific scenarios – consistently showing significant OOD generalization improvements require discovery of casual relationships. Accurate discovery of causal relationships would not only improve accuracy and reliability, but also enable explainable decision making, which is crucial for high-stakes applications such as healthcare or finance (Shin, 2021).

Learning the true causal relationships is very challenging, fundamentally (Schölkopf, 2022). It is infeasible to consider all combinations for factors of variation (such as shape, size and color of an image), as it would be exponential in size ($N^M$ combinations with $M$ categorical features where each can take $N$ different values). Effective methods should reduce the prohibitively-high search cost and data inefficiency while accurately discovering the underlying mechanisms. Causal discovery has been studied using various approaches. There are methods based on interventional experiments by randomized controlled trials (Pearl, 2009), but they are often prohibitively costly. A more realistic setting is learning from observational data. Constraint-based algorithms(Spirtes et al., 2000; 2013) directly conduct independence tests to detect causal structure. Score-based algorithms (Chickering, 2002; Huang et al., 2018) adopt score functions consistent with the conditional independence statistics, however, these can only find the Markov-equivalence class (Guo et al.,

2020). Functional causal models (Shimizu et al., 2006; Peters et al., 2014) aim to identify the causal structure from the equivalence class, but the heuristic directed acyclic graph (DAG) search methods suffer from high computational cost and local optimality, as the number of nodes increases. To address this problem, NOTEARS (Zheng et al., 2018) proposes a differentiable optimization framework. NOTEARS-MLP (Zheng et al., 2020) and Gran-DAG (Lachapelle et al., 2019) extend NOTEARS to non-linear modeling with DNNs. NoFear (Wei et al., 2020) reevaluates NOTEARS continuous optimization framework, and subsequently, innovates a local search algorithm that enhances its performance. GOLEM (Ng et al., 2020) propose a likelihood-based structure learning method that applies soft sparcity and DAG constraints. DARING (He et al., 2021) uses constraints on independent residuals to facilitate DAG learning. One common shortcoming of these approaches is relying on empirical risk minimization. Recent work has shown on the other hand that invariant risk minimization (Arjovsky et al., 2019) can be very powerful in preventing the absorption of spurious correlations during DAG learning.

Our goal in this paper is to push the state-of-the-art in accurate discovery of structural causal models (SCM), and as a consequence, improve the accuracy and reliability of models especially in the presence of severe distribution shifts. Motivated by the limitations of existing work mentioned above, we propose Invariant Structure Learning (ISL), a framework that yields causal explainability, based on tying generalization and SCM learning. Intuitively, better generalization should lead to more accurate SCM learning, and an accurate causal structure should yield improved robustness and generalization. ISL encourages reinforcement between these two goals. Specifically, ISL uses generalization accuracy as a constraint to learn the invariant SCM (as a DAG) that represents the causal relationship among variables. Take Fig. 1 as an example, where we simplify the object recognition task by using variables to represent the key factors: $X1$: object shape, $X2$: object texture (including color), $X3$: image background (as context), with the output label $Y$. Fig. 1 (a) shows the ground truth (GT) Structural Causal Model (SCM). During training, the data (Fig. 1 (b)) consist of samples from different environments. Baseline methods such as NOTEARS-MLP and CASTLE directly estimate the underlying causal structure, which leads to spurious correlations being absorbed, which in turn results in sub-optimal test accuracy. Our method ISL, on the other hand, learns the invariant structure that correctly identifies the SCM and yields better test accuracy. Overall, our contributions are highlighted as:

- We propose Invariant Structure Learning (ISL), a novel learning framework that yields accurate causal explanations by mining the invariant causal structure underlying the training data, and generalizes well to unknown out-of-distribution test data.
- We generalize ISL to self-supervised causal structure learning, which first treats the discovered invariant correlations as potential causal edges, and then uses a DAG constraint to finalize the causal structure.
- We demonstrate the effectiveness of ISL on various synthetic and real-world datasets. ISL yields state-of-the-art SCM discovery (clearly outperforming alternatives on real-world data) with a particularly prominent improvement for complex graphs structures. In addition, ISL improves the test prediction accuracy throughout, with especially large improvements in cases with significant data drifts (up to $\sim 80\%$ MSE reduction compared to alternatives).

## 2 Related Works

**Improving machine learning generalization.** Many different approaches have been studied to improve generalization (i.e. bringing the test performance closer to training). Regularization methods (Arpit et al., 2017; Ng, 2004; Tibshirani, 1996; Hoerl & Kennard, 1970; Hinton et al., 2012; Wager et al., 2013), early stopping (Goodfellow et al., 2016), gradient clipping (Pascanu et al., 2013), batch normalization (Ioffe & Szegedy, 2015), data augmentation (Yaeger et al., 1996; Krizhevsky et al., 2012) are among the most popular ones. These aren't based on discovering the true relationship between the features. Towards generalization improvements with input feature discovery direction, supervised auto-encoders (Le et al., 2018) add a reconstruction loss for the input features as a regularizer. Recently, some works (Janzing, 2019; Bahadori et al., 2017) combine causal discovery with model regularization for better generalization. CASTLE (Kyono et al., 2020) implicitly uses underlying Structural Equation Model (SEM) reconstruction as the regularization to improve model generalization. However, it can't explicitly yield a DAG for the causal structure and it can't completely prevent learning spurious correlations. ISL addresses these two challenges by learning the

invariant structure across environments and outputting a DAG to describe the causal data structure, and eventually showing better generalization.

**Causal structure discovery.** Constraint-based causal discovery algorithms (Spirtes et al., 2000; 2013), and some score-based methods(Chickering, 2002; Huang et al., 2018; Shimizu et al., 2006; Peters et al., 2014) conduct exhaustive and heuristic search for the DAG structure, yielding combinatorial explosion issue when they scale up to a larger number of the nodes. NOTEARS (Zheng et al., 2018) proposes directly applying a standard numerical solver for constrained optimization to achieve a global approximate solution overcoming the scalability bottleneck. They formulate the structure learning problem as maximum likelihood estimation over observational data with the additional constraint that the weight matrix has to represent a DAG with the acyclicity and sparsity properties. NOTEARS-MLP (Zheng et al., 2020) and Gran-DAG (Lachapelle et al., 2019) extend NOTEARS to non-linear functions by using DNNs. RL-BIC (Zhu et al., 2019) uses Reinforcement Learning to search for the DAG with the best scoring. GOLEM (Ng et al., 2020) applies a likelihood-based objective with soft sparsity and DAG constraints. These methods don't consider using the generalization quantification as an indication or constrain during DAG learning, which makes the learned DAG sometimes absorb biases and spurious correlations from data.

**Invariant Learning.** The field of invariant learning is increasingly gaining traction, largely due to its implications for causality. EIIL (Creager et al., 2021) focuses on the inference of environments to facilitate invariant learning, while 'ZIN' (Lin et al., 2022) delves into the conditional feasibility and methods for environment inference. Another noteworthy contribution GALA (Chen et al., 2023), which advocates for invariant learning without the need for explicit environment partitioning. Despite these advances, our work stands apart in its capability for causal discovery. Unlike the aforementioned studies, our methodology uniquely allows for simultaneous learning of causal structures and predictions.

## 3  Methodology

In Section 3.1, we first present the problem definition and the motivation behind our work by discussing how spurious correlation can affect the model generalization and how its influence can be alleviated. Section 3.2 describes the proposed ISL framework for a supervised learning setting. In Section 3.3, we extend our discussion to the generalization of our approach in a self-supervised setting.

### 3.1  Motivations

**Problem definition.** Standard supervised learning is defined for a dataset with given input variables $\hat{X} = (Y, X_1, X_2, ... X_d)$, including $\mathbf{X} = \{X_i\}_{i=1}^d \in \mathcal{X}$ and $Y \in \mathcal{Y}$, the goal is to learn a predictive model $f_Y : \mathcal{X} \rightarrow \mathcal{Y}$. $P_{\mathbf{X},Y}$ denotes the joint distribution of the features and target, $\mathcal{D}_{train}$ denotes the training data with $N$ samples, and $\mathcal{D}_{test}$ denotes the testing data. Ideally, we expect both $\mathcal{D}_{train}$ and $\mathcal{D}_{test}$ to be i.i.d., sampled from the same distribution $P_{\mathbf{X},Y}$. However, it is hard to satisfy this condition for real-world data. It becomes more severe when the model overfits to the training set or the training set does not reveal the underlying distribution $P_{\mathbf{X},Y}$.

**Spurious correlations and causality.** One perspective to explain poor generalization due to overfitting is models learning spurious correlations. Broadly, a correlation can be considered as spurious when the relationship does not hold across all samples in the same manner (Arjovsky et al., 2019). For example, for the image recognition task in Fig. 1(a), the model may use the green color of the grass to recognize cows, instead of complete profile of its shape. The correlation between green-colored grass and the cow label would be spurious and not consistent. In contrast, with causal learning, our goal is to learn stable and invariant relationships, that generalize well. Let's consider that there is a SCM defining how the random variables $\hat{X} = (Y, X_1, X_2, ... X_d)$ define each other. The target variable $Y$ is generated by a function $f_Y(Pa(Y), u_Y)$, where $Pa(Y)$ denotes the causal parents of $Y$ in SCM. Non-parametric SEM (Pearl et al., 2000) proves that if we use the causal parents of $Y$ as inputs to predict $Y$, the learned model would be optimal and generalize well on the unknown test set. Identifying causal parents of $Y$ from spurious correlations is the key to obtain better generalization and causal explainability.

**Invariant structure across environments:** An environment is used to distinguish different properties of data (such as the generative source characteristics), and can help reveal reasons for spurious correlations. Examples of environments can be different devices for capturing the images, or the hospitals at which the patient data are collected. Broadly, it can be considered as a set of conditions, interpreted as 'context' of the data (Moneda, 2021). As an important indication of distinguishing causality from spurious correlations, the causal structure of $Y$ should be invariant across all possible environments. Our goal is to learn such invariant structure across environments, which should yield better generalization.

## 3.2 Learning framework

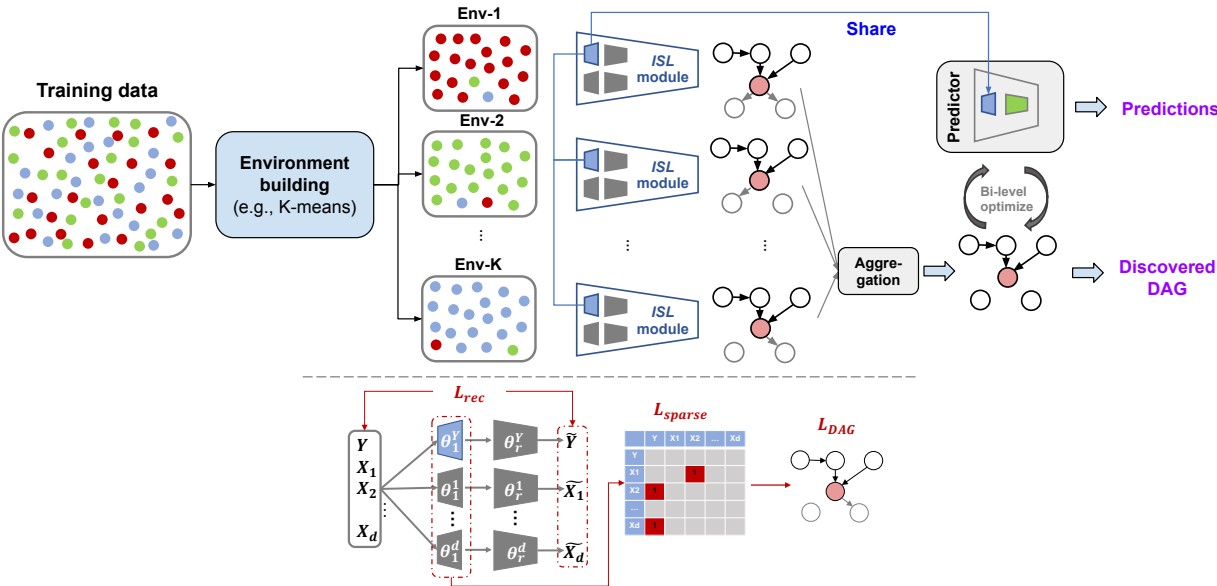

Figure 2: **Top:** The proposed Invariant Structure Learning (ISL) framework. Given raw data, we build different environments using unsupervised clustering, unless the data source information is provided. For different environments, each ISL module outputs a summarized DAG to represent the learned invariant structure. An aggregation mechanism then selects the optimal predictor based on a graph structure that reflects the causal mechanisms in the data more accurately. During training, the constraint on the $Y$ prediction across environments helps learning an invariant structure. Consequently, the learned DAG leads to a superior predictor. **Bottom:** Details of the ISL module. $\theta_1^Y$ is the invariant structure of $Pa(Y)$ shared across all modules.

Fig. 2 shows the overall Invariant Structure Learning (ISL) framework. Given raw training data, we first build different environments. If we had the information on data generation and collection process, it would be simple to build different environments based on them. Without such information, we propose to build environments in an unsupervised way using unsupervised clustering, K-means (Lloyd, 1982), which clusters the raw data into different clusters that each representing an environment. The clustering can be done for the raw data or learned representations, which would require an encoder to map raw data. To determine the number of clusters, we use Elbow (Thorndike, 1953) and Silhouette (Rousseeuw, 1987) methods. To balance the data size across environments, we employ upsampling. We augment the data size to reach to $n$, the largest number of data samples in an environment. Fig. 2(top) depicts the environment building process, where each color represents a different source or generation method for the sample. In general, at least two diverse environments are sufficient to learn the invariant structure (Arjovsky et al., 2019). We show that ISL is robust to the number of environments and typically an intermediate value is optimal.

After assigning data samples to different environments, our goal is to learn the invariant structure that results in superior generalization for predicting $Y$. In each environment, using that environment's data, an ISL module (see Fig. 2(bottom)) independently learns a DAG which defines the variable relationship for that specific environment. To learn the invariant structure for predicting $Y$, we add a constraint among ISL

modules that the parameters to reconstruct $Y$ should be identical across environments. The desiderata for invariance is expressed as a loss function over all environments in the training data. Ideally, the generalization goal would be minimization of an OOD risk $R^{OOD}(f) = \max_{e \in \epsilon_{all}} R^e(f(X))$ over all possible environments, not only the ones in the training data. We aim to approximate this with a tractable objective. We propose the risk under a certain environment $e$ as $R^e(f) = \mathbb{E}_{X^e, Y^e}[l(f(X^e), Y^e))]$, where $l$ denotes the loss function. We decompose the objective function $f()$ into two components. The first is to find the representation of causal parents of $Y$ from given $X$, which can be considered as the invariant structure for $Y$ across environments. The second is to optimize the classifier with the learned $Pa(Y)$ as the input. $\theta$ is a multi-layer perceptron used to approximate $f()$. It consists of: (i) $g(\cdot)$ with parameters $\theta_1^Y$ to learn a representation of $Pa(Y)$, and (ii) $h(\cdot)$ that inputs the representation and yields the prediction for $Y$. To learn the representation of $Pa(Y)$, $g(\cdot)$ should follow the causal structure of $Y$. Overall, the proposed objective to learn the invariant causal structure (as a DAG) is summarized as below:

$$\min_{\theta_1^Y, h} \sum_{e \in \epsilon_{all}} R^e(h \circ \theta_1^Y(X)),$$
$$s.t. \quad \theta_1^Y, \theta_r^Y, \theta^X = \arg\min_{\theta} \sum_{e \in \epsilon_{all}} \mathcal{R}_{DAG}(\hat{X}, \theta). \tag{1}$$

This objective is in bi-level optimization form. The outer loop is for the final goal of obtaining the predictive model for $Y$ to generalize well on all environments, requiring that $\theta_1^Y$ extract the representation of $Pa(Y)$. The inner loop adds the constraint that $\theta_1^Y$ should be the invariant across all the environment during learning their DAGs. As shown in Fig. 2, we use a DNN to learn the SCM (represented as a DAG) of the dataset. The parameter of $\theta$ consists of three parts: first layer to reconstruct variable Y, $\theta_1^Y$, rest layers to reconstruct variable Y, $\theta_r^Y$, and layers to reconstruct other variables X, $\theta^X$. We use the following objective function represented as the DAG loss $\mathcal{R}_{DAG}(\hat{X}, \theta)$:

$$\mathcal{R}_{DAG}(\hat{X}, \theta) = \mathcal{L}_{rec}(\hat{X}, \theta) + \frac{\rho}{2}|h(W)|^2 + \alpha h(W) + \beta \mathcal{L}_{sparse}(\theta), \tag{2}$$

where $\mathcal{L}_{rec}(\theta) = \frac{1}{2N}||\hat{X} - \theta(\hat{X})||_F^2$ and $h(W) = \text{Tr}(e^{W \odot W}) - d$, with $|| \cdot ||_F$ being the Frobenius norm, $\mathcal{L}_{DAG} = \frac{\rho}{2}|h(W)|^2 + \alpha h(W)$ denotes the constrain of DAG. $N$ denotes the number of samples, $\text{Tr}()$ is the trace operator, $W$ is a $(d+1) \times (d+1)$ adjacency matrix ($W \in \mathbb{R}^{(d+1) \times (d+1)}$) which represent the connection strength between variables and the final DAG is summarized from $W$. (Zheng et al., 2018) proves that $W$ is a DAG if and only if $h(W) = 0$. $W$ is summarized from the first layer $\theta_1$ of $\theta$. Specifically, $[W]_{k,j}$ is the $L_2$ norm of the $k$-th row of the parameter matrix $\theta_1^j$. $\theta^j$ is the DNN parameters used to reconstruct variable $j$, that we decompose as the first layer $\theta_1^j$ and the remaining layers $\theta_r^j$ (Fig. 2). $\mathcal{L}_{sparse}(\theta) = \beta_1||\theta_1^Y||_1 + \beta_2||\theta_r^Y||_2 + \beta_3||\theta^X||_1 + \beta_4||\theta^X||_2$, where $|| \cdot ||_1$ and $|| \cdot ||_2$ denote $l_1$ and $l_2$ regularization, respectively, and $\beta_i$ are hyperparameters that can be optimized on validation set. Eq. 2 is a solution using Augmented Lagrangian (Fortin & Glowinski, 2000) approach, where $\alpha > 0$ and $\rho > 0$ are gradually increased to find solutions that minimize $h(W)$. To learn the invariant structure across different environments, in all modules, we use a *shared* layer $\theta_1^Y$. It learns a representation of $Pa(Y)$ given input feature $X$, ($\theta_1^Y(X) \approx Pa(X)$), which is a constraint to learn the invariant structure of $Y$ prediction among environments. We simplify the training of Eq. 1 and the overall training objective of the proposed ISL is defined as:

$$\min_{\theta, h} \sum_{e \in \epsilon_{all}}^n (R^e(h \circ \theta_1^Y(X)) + \gamma \mathcal{R}_{DAG}^e(\hat{X}, \theta_1^Y, \theta_r^Y, \theta^X)), \tag{3}$$

where $\gamma$ is the trade-off parameter and $\mathcal{R}_{DAG}^e(\hat{X}, \theta_1^Y, \theta_r^Y, \theta^X)$ is the invariant structure constraint. We propose solving this problem with a second order Newton method, L-BFGS-B (Zhu et al., 1997).

Algorithm 1 summarizes the training procedure. After DAG learning converges at all environments, we obtain the invariant structure of $Y$ prediction by first computing $Y$-related columns in adjacency matrix $W$ from shared $\theta_1^Y$, and then using a threshold (please refer to Appendix B for details on how the threshold was selected) select the learned $Pa(Y)$ ($Y$-related DAG). For the final target, we fix all parameters in Eq. 3 except $h(\cdot)$, and fine-tune $h(\cdot)$. To obtain the overall DAG for the entire dataset, we aggregate the DAG across different environments by keeping only the overlapping edges across all environments. Please note that $h(\cdot)$ always takes the output of $g(\cdot)$ as input and yields the prediction for $Y$. In practice, the parameters of $h(\cdot)$

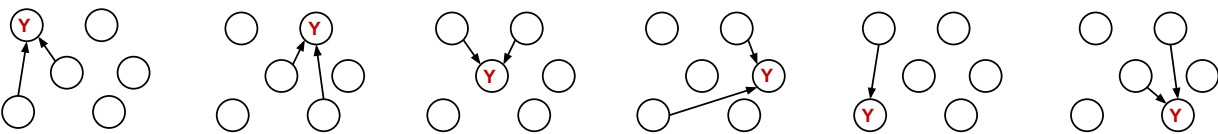

(a) **Step 1:** Iteratively set each variable as target and propose an invariant structure for each variable.

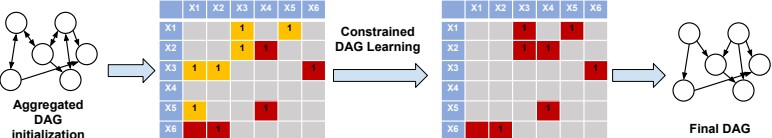

(b) **Step 2:** Aggregate all proposed causal parents for each variable into a single graph (may not be a DAG), then adjust the activation ability of DNN by de-activating all parameters that are related to the non-proposed edges, and lastly optimize Eq. 2 to obtain final DAG.

Figure 3: ISL in self-supervised setting.

are updated twice: initially, $g(\cdot)$ and $h(\cdot)$ are combined to discover the causal parents of the target $Y$ through the learning of $g(\cdot)$, and $h(\cdot)$ merely assisting the learning of $g(\cdot)$ through the DAG constraint and ERM. $h(\cdot)$ maybe suboptimal for $Y$ prediction at this stage. Then, after determining the causal parent of the target variable $Y$ by applying a threshold in the weight matrix of $W$ and reconstructing the DAG, we updated $g(\cdot)$ and fix it. Then, we fine-tune $h(\cdot)$ using only the causal parent variables discovered by $g(\cdot)$(fixed) to predict the target variable $Y$.

---

**Algorithm 1:** Supervised Invariant Structure Learning

**Input:** Dataset $\mathcal{D}$
**Output:** DAG, $Y$ predictor $f(X) = h \circ \theta_1^Y(X)$
1 Build $n$ environments $\epsilon_{all}$, for each $e \in \epsilon_{all}$, $\rho^e = 1$, $\alpha^e = 0$, $h^e(W) = \infty$.
2 Termination conditions $h(W)_{tol} = 10^{-8}$, $\rho_{max} = 10^{16}$, max iteration as $N_{MAX}$ with $i = 0$.
3 **while** $i < N_{MAX} = 100$ $and$ $\max_{e \in \epsilon_{all}}(h^e(W)) > h(W)_{tol}$ $and$ $\min_{e \in \epsilon_{all}}(\rho^e) < \rho_{max}$ **do**
  $\quad$ i += 1
4 $\quad$ **for** $e = 1$ to $k$ **do**
  $\qquad$ Calculate $R^e(h \circ \theta_1^Y(X))$ and $\mathcal{R}_{DAG}^e(\hat{X}, \theta_1^Y, \theta_r^Y, \theta^X)$ in Eq. 2
  $\qquad$ Update $h, \theta_1^Y, \theta_r^Y, \theta_r^Y, \theta^X$ with L-BFGS-B (Zhu et al., 1997);
  $\qquad$ Calculate $W$ from $\theta_1^Y$; Update $h^e(W)$; Update $\rho^e$ and $\alpha^e$
5 Summarize DAG from $\theta_1$
6 Fix all trainable parameters in Eq. 3 except $h$, fine-tune $h$ and obtain final $f(X) = h \circ \theta_1^Y(X)$.

---

### 3.3 Generalizing to self-supervised setting

In many scenarios, the target labels aren't available, rendering self-supervised causal graph discovery as an paramount problem. Conventional functional causal models, such as NOTEARS, aim to find a trade-off between three objectives, which are optimal to SEM: $X = XW$, whilst $W$ should both resemble a DAG and be sparse (see Eq. (3)). As all variables aren't distinguishable with equal importance, there is no prior knowledge about which nodes should be the source or target nodes. Due to the large variance among node distribution caused by variable semantic meaning, reconstruction accuracy driven learning is unstable, and sensitive to the variable distribution – some variables can be described using a simple distribution, while others may be hard to estimate due to the differences in data source. It can lead to a local minima, causing the learned DAG deviate from the real causal structure (Kaiser & Sipos, 2021). We propose a two-step DAG learning approach, as shown in Fig. 3.

**Step 1. Invariant causality proposal:** We first build multiple environments. Then, we iteratively set each node as $Y$ (Fig. 3) and run ISL to propose the invariant structure for $Y$ as candidate causal parents.

ISL keeps the invariant variables that are important to prediction of $Y$ under the overall DAG constrain. As such, the learned invariant structure corresponds to either true causal parents of $Y$ or the variables which have strong correlation with $Y$, thus treated as candidate of causal parents of $Y$).

**Step 2. Constrained graph discovery:** We aggregate the candidate causal parents of each variable in Step 1 and form an aggregated graph (Fig. 3), where there are bi-direction edges, which isn't allowed in a DAG. We can build a $(d+1) \times (d+1)$ binary adjacency matrix $W$ to represent the graph where $d+1$ represent the number of nodes. $j$-th column of $W$ represent the potential causal parents of node $j$. As described in Sec. 3.2, during DAG learning, for each variable $X_j$, we use a DNN $\theta^j$ to reconstruct $X_j$ given other variables. There is a corresponding mapping between the $j$-th column of $W$ and the first layer $\theta_1^j$ of DNN $\theta^j$: the $k$-th row in the parameter matrix of $\theta_1^j$ encode the contribution of node $k$ to node $j$, which associate with the value of $[W_{k,j}]$. To narrow down the search space and improve DAG learning, if $[W_{k,j}]$ is 0 (node $k$ aren't potential causal parent of node $j$ summarized from Step 1), we deactivate the corresponding parameter by fixing the value of $k$-th row in $\theta_1^j$ as 0. if $[W_{k,j}]$ is 1, we don't add weight constraint to the first layer $\theta_1^j$ in the DAG. We use the parameter modification as a constraint on DAG learning and run a constrained version of DAG learning (Eq. 2) to obtain the final DAG.

---

**Algorithm 2:** Self-Supervised Invariant Structure Learning

---

**Input:** Dataset $\mathcal{D}$
**Output:** DAG
   # Step 1 Invariant causality proposal
**1** Build $n$ environments, $\epsilon_{all}$
**2 for** $X_i = X_1$ *to* $X_d$ **do**
   | Set $Y = X_i$ and run algorithm 1 (main manuscript), select only the $Pa(X_i)$
   # Step 2 Constrained graph discovery
**3** Aggregate $Pa(X_i)$ for each variable to form the initial graph $G$ (may not be a DAG).
**4** Summarize a initial adjacency matrix $W'$
**5** Add weight constraint on $\theta$ based on $W'$
**6** Run DAG mining (Eq.2 in main manuscript) to optimize $\theta$ and yield DAG to describe the approximated causal structure.

---

## 4 Experiments

In this section, we evaluate the proposed ISL framework for causal explainability and better generalization. We conduct extensive experiments in two settings based on the availability of target labels: supervised learning tasks in Sec. 4.1 and self-supervised learning tasks in Sec. 4.2. Details and more results are provided in the Appendix D.

**Baselines:** On causal explainability, we choose NOTEARS-MLP (Zheng et al., 2020), GOLEM (Ng et al., 2020), and NoFear (Wei et al., 2020) as the baselines for learning the SCM which represented as a DAG. On target prediction, we choose a standard MLP and CASTLE (Kyono et al., 2020) as the baseline methods.

**Metrics:** We evaluate the estimated Y-related DAG and whole DAG structure using Structural Hamming Distance (SHD): the number of missing, falsely detected or reversed edges, lower the better. We evaluate the target ($Y$) prediction accuracy in Mean Squared Error (MSE). We compute SHD and the errors for multiple times and report the mean value.

### 4.1 Supervised learning tasks

### 4.1.1 Synthetic data

We first examine the performance of ISL in accurately discovering the casual structure, as well as the target prediction performance using synthetic tabular datasets with known casual structure information as well as the target labels. We aim to mimic challenging scenarios encountered for data generation and collection

Table 1: Synthetic tabular data experiments in supervised learning setting. Note that black-box MLP and CASTLE can't provide DAGs. ISL yields lower MSE for ID and OOD, and lower SHD.

| Number of nodes | Metrics | MLP | NOTEARS-MLP | CASTLE | GOLEM | NoFear | ISL (Ours) |
|---|---|---|---|---|---|---|---|
| 3 (c=2, s=1) | ID MSE | 0.008 | 0.090 | 0.012 | 0.171 | 0.422 | **0.005** |
| | OOD MSE | 0.016 | 0.191 | 0.020 | 0.394 | 0.451 | **0.010** |
| | Average SHD | - | 2 | - | 3 | 2 | **0** |
| 4 (c=2, s=2) | ID MSE | 0.006 | 0.082 | 0.019 | 0.250 | 0.441 | **0.006** |
| | OOD MSE | 0.014 | 0.152 | 0.032 | 0.250 | 0.411 | **0.009** |
| | Average SHD | - | 2 | - | 4 | 2 | **0** |
| 5 (c=3, s=2) | ID MSE | 0.004 | 0.093 | 0.020 | 0.250 | 0.427 | **0.004** |
| | OOD MSE | 0.004 | 0.060 | 0.016 | 0.250 | 0.419 | **0.004** |
| | Average SHD | - | 3 | - | 5 | 3 | **0** |
| 9 (c=4, s=5) | ID MSE | 0.006 | 0.061 | 0.025 | 0.250 | 0.416 | **0.005** |
| | OOD MSE | 0.031 | 0.174 | 0.160 | 0.250 | 0.407 | **0.005** |
| | Average SHD | - | 4 | - | 10 | 4 | **0** |
| 20(c=10, s=10) | ID MSE | 0.008 | 0.051 | 0.137 | 0.250 | 0.403 | **0.008** |
| | OOD MSE | 0.018 | 0.251 | 0.252 | 0.250 | 0.434 | **0.007** |
| | Average SHD | - | 9 | - | 19 | 9 | **1** |

processes in real world, that the data may consist of samples from different environment sources, while the target-related causal structure is consistent in the entire dataset (e.g., Fig. 1). We construct the synthetic datasets in the following way (more details are provided in Appendix A):

- **Step 1:** We randomly sample an initial DAG $G'$ following Erdos-Renyi or Scale-Free schema with different edge densities. We randomly select one node (which isn't the source node) as the target $Y$. We calculate the number of causal parent nodes $C$ of $Y$, $c$. If $c < c_{min}$, we randomly add $c_{min} - c$ number of nodes into $C$ as the causal parents of $Y$.
- **Step 2:** To simulate the spurious correlations, we create $s \in [1, ...k]$ new nodes $S$, and these nodes act as causal descendants of $Y$. After defining the causal parents and descendants of $Y$, now we obtain the GT DAG $G$. For all nodes $X$ except $Y$ and $S$, we define an ANM $X = F(X) + \epsilon$ to generate data on top of $G$, where $F$ is a two-layer MLP whose parameters are uniformly sampled from (-2, -0.5) $\cup$ (0.5, 2). $\epsilon$ is the external noise which is randomly sampled from Gaussian, Exponential and Uniform. $Y$ is generated from its causal parents $C$, $Y \sim P(Y = 1|\text{sigmoid}(G(C) + \epsilon))$. $G$ can be either linear (uniformly random weight matrix) or non-linear (same initialization method as $F$).
- **Step 3:** We randomly select the number of environments $e$ from the uniform distribution of [2, 5]. For each environment, all nodes (except for $s$ added spurious correlation nodes $S$) in the GT DAG $G$ follows the ANM (defined in Step 2) but with different random seed and noise term. For $S$, their correlation to $Y$ isn't invariant among environments and controlled by a continuous variable $r \in [0, 1]$. Specifically, for each node $S_i$ in $S$, $S \sim P(S = 1|Y = 1) = r = P(S = 0|Y = 0)$.
- **Step 4:** We generate two different kinds of test set: In-distribution (ID) and Out-of-distribution (OOD). Both have the same number of environments with the training set. ID test set uses the same value of $r$ as training set, while OOD test set uses uniformly random sampled $r$, which represents the unknown test environments. $S \sim P(S = \hat{g}(Y)|Y) = r$, $P(S \sim \text{random variable}) = 1\text{-}r$.

We generate different sizes of graphs with 5 $c$ and $s$ combinations (see Table. 1) (10 datasets with 1000 samples for each environment). Table 1 shows that ISL significantly outperforms others for both $Y$ prediction and $Y$-related DAG learning. Particularly in OOD scenarios, the outperformance is more prominent, up to 83% decrease in MSE compared to black-box MLP and 96% decrease compared to CASTLE. This is attributed to more accurate casual graph discovery (evident from lower SHD), allowing to capture dynamics that are consistent between ID and OOD data, and hence the model generalizes better.

**Counterfactual simulations:** Besides the accurate predictions for test data, causal structure discovery is also notable for its capability of accurate modeling of counterfactual outcomes, i.e. predicting how the output would change with certain input changes. To demonstrate this, we design experiments by modifying the dataset used above. We randomly select a node $X_i$ from causal parents set $C$ or spurious correlation set $S$ and change the value of $X_i$ while keeping the other nodes unmodified. Then, we test the prediction accuracy on this dataset. Table. 2 shows that ISL yields more accurate counterfactual outcomes, compared to alternatives. Particularly when the counterfactual source is spurious correlation variables, baseline methods are much worse at outcome predictions.

Table 2: Synthetic tabular data counterfactual simulation experiments. MSE is shown for various counterfactual outcomes, obtained by modifying the 'counterfactual source' variables.

| Counterfactual source | MLP | NOTEARS-MLP | CASTLE | ISL (Ours) |
|---|---|---|---|---|
| Causal parent X1 | 0.021 | 0.280 | 0.034 | **0.016** |
| Causal parent X2 | 0.043 | 0.301 | 0.064 | **0.012** |
| Spurious correlation S1 | 0.184 | 30.962 | 0.471 | **0.012** |

### 4.1.2 Real-world data

We perform supervised learning experiments on real-world datasets with GT causal structure: Boston Housing (Binder et al., 1997; bos) and Insurance (Binder et al., 1997; ins) datasets. For each, we randomly split the train/validation/test with the proportion 0.8/0.1/0.1. We conduct three experiments and show the average performance. We consider the accuracy for $Y$ prediction and target-related DAG (causal parents of $Y$) learning. Specifically, Boston Housing contains information collected by the U.S Census Service concerning housing in Boston. There are 14 attributes including 1 binary variable and 506 samples in which the median value of homes (MED) is to be predicted. For ISL, we first calculate the Within-Cluster-Sum of Squared (WSS) errors for different values of $k$, and choose the $k$ for which the WSS starts to diminish. Based on this, we build $k = 2$ environments. We obtain the $Y$-related GT DAG of Boston Housing from (Wei & Feng, 2021; Zhang et al., 2012). The Insurance dataset is based on a network for car insurance risk estimation. The network has 27 nodes and 52 edges with 20000 samples. The Insurance dataset provides the GT causal structure as a DAG. Three of the observable nodes ('PropCost', 'LiabilityCost' and 'MedCost') are designated as 'outputs'. Besides the designated output, we add other variables 'CarValue' (based on the importance for the task) as the target as well. For ISL, similarly, we use K-means clustering to build $k = 3$ different environments. Table. 4 summarizes the results for $Y$ prediction and $Y$-related causal structure learning. We observe that ISL significantly outperforms black-box MLP in all cases (up to 74% MSE reduction), as well as NOTEARS-MLP and CASTLE.

Table 3: Supervised learning experiments on real-world data. Note that MLP and CASTLE cannot provide DAGs (and thus don't have SHD).

| Dataset | Target | Metrics | MLP | NOTEARS-MLP | CASTLE | GOLEM | NoFear | ISL (Ours) |
|---|---|---|---|---|---|---|---|---|
| Boston Housing | MED | MSE (↓) | 0.16 | 0.12 | 0.10 | 0.53 | 0.53 | **0.05** |
| | | SHD (↓) | - | 2 | - | 3 | 3 | **1** |
| Insurance | 'PropCost' | MSE (↓) | 0.40 | 0.99 | 0.36 | 0.68 | 1.03 | **0.34** |
| | | SHD (↓) | - | 2 | - | 1 | 1 | **0** |
| | 'MedCost' | MSE (↓) | 0.69 | 1.03 | 0.55 | 0.86 | 0.99 | **0.52** |
| | | SHD (↓) | - | 2 | - | 4 | 4 | **0** |
| | 'LiabilityCost' | MSE (↓) | 0.94 | 0.39 | 0.38 | 0.50 | 1.03 | **0.25** |
| | | SHD (↓) | - | 1 | - | 2 | 3 | **0** |
| | 'CarValue' | MSE (↓) | 0.23 | 0.60 | **0.23** | 0.97 | 0.97 | **0.23** |
| | | SHD (↓) | - | 2 | - | 6 | 4 | **1** |

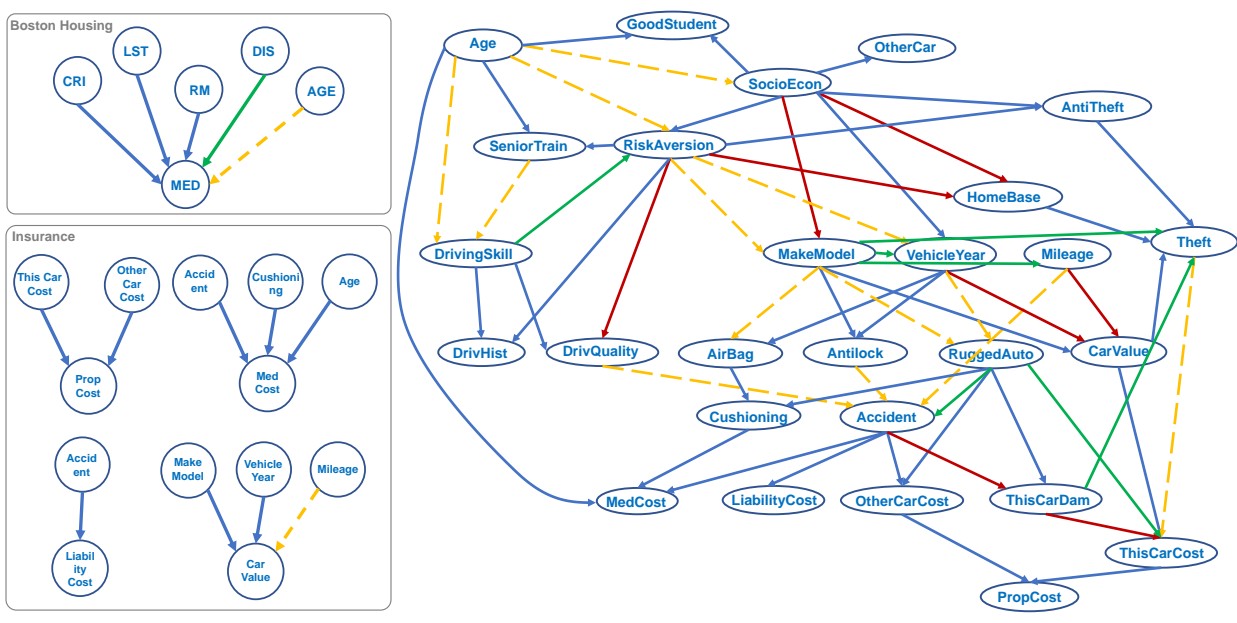

(a) Y-related DAG (Supervised)      (b) Discovered DAG for Insurance dataset (Self-supervised)

Figure 4: Visualization of discovered causal structure in (a) supervised and (b) self-supervised settings. Blue solid arrows are overlapped edges between our results and GT, red solid arrow denotes the edges that we can identify but with wrong direction, green solid arrow denotes our proposed edge that is not contained for, yellow dash arrows denotes our missing edges that GT contains.

**Impact of the number of environments:** The default number of environments we used in all experiments is 3. We investigate the impact of the number of environments on Boston Housing and synthetic data. Table 4 shows the clear value of having multiple environments – with only one, the invariant constraint is not effective, yielding worse results. Increasing the number of environments has diminishing returns. More ablation studies are provided in the Appendix C.

Table 4: The impact of number of environments for ISL in supervised learning setting.

| Dataset | Target | Metrics | ISL (e=1) | ISL (e=2) | ISL (e=7) |
|---------|--------|---------|-----------|-----------|-----------|
| Boston Housing | MED | MSE | 0.067 | 0.051 | 0.051 |
|  |  | SHD | 5 | 1 | 1 |
| Synthetic data (Fig. 1) | Y | MSE | 0.110 | 0.022 | 0.020 |
|  |  | SHD | 2 | 0 | 0 |

## 4.2 Self-supervised learning

For self-supervised learning tasks, there is no target variable, and the goal is to learn accurate SCM, represented as a DAG, that represent the underlying causal structure of given dataset. We conduct experiments on two real-world datasets: Sachs (Sachs et al., 2005; sac) and Insurance (Binder et al., 1997; ins) datasets. The Sachs dataset is for the discovery of protein signaling network on expression levels of different proteins and phospholipids in human cells (Sachs et al., 2005), and is a popular benchmark for causal graph discovery, containing both observational and interventional data. The true causal graph from (Sachs et al., 2005) contains 11 nodes and 17 edges. We conduct our two-stage DAG learning based on ISL by building 3 environments and compare the DAG results with different baselines. Table. 5 shows that ISL outperforms all other methods in correct discovery of the GT DAG on both Sachs and Insurance. On the challenging Insurance data, the number of corrected edges is 72% higher for ISL, compared to NOTEARS-MLP.

Table 5: Self-supervised causal graph discovery on the Sachs and Insurance datasets.

| Dataset | Method | Total Edges | Correct Edges ($\uparrow$) | SHD ($\downarrow$) |
|---------|--------|-------------|----------------------------|---------------------|
| Sachs | RL-BIC (Zhu et al., 2019) | 10 | 7 | 11 |
| | GraN-DAG (Lachapelle et al., 2019) | 10 | 5 | 13 |
| | NOTEARS-MLP (Zheng et al., 2020) | 11 | 6 | 11 |
| | DAG-GNN (Yu et al., 2019) | 15 | 6 | 16 |
| | GOLEM (Ng et al., 2020) | 11 | 6 | 14 |
| | NOTEARS (Zheng et al., 2018) | 20 | 6 | 19 |
| | ICA-LiNGAM (Shimizu et al., 2006) | 8 | 4 | 14 |
| | CAM (Glymour et al., 2019) | 10 | 6 | 12 |
| | DARING (He et al., 2021) | 15 | 7 | 11 |
| | ISL (Ours) | 12 | **8** | **8** |
| Insurance | NOTEARS-MLP (Zheng et al., 2020) | 35 | 18 | 39 |
| | NOTEARS (Zheng et al., 2018) | 24 | 10 | 46 |
| | GOLEM (Ng et al., 2020) | 36 | 28 | 61 |
| | NoFear (Wei et al., 2020) | 15 | 10 | 49 |
| | ISL (Ours) | 46 | **31** | **27** |

## 5 Conclusions

We propose a novel method, ISL, for accurate causal structure discovery. The ISL framework is based on splitting the training data into different environments and learning the structure that is invariant to the selected target. We demonstrate the effectiveness of ISL in both supervised and self-supervised learning settings. On synthetic and real-world datasets, we show that ISL yields more accurate causal structure discovery compared to alternatives, which also results in superior generalization, especially against severe distribution shifts.

## 6 Limitations and Future Work

Our approach has proven effective in uncovering the causal structure by leveraging the assumption of its invariance across multiple environments. However, this method relies heavily on partitioning the dataset into distinct environments via clustering, a process that may present challenges in cases where data is characterized by high dimensionality or fluctuating densities.

Therefore, a potential limitation of our current method is its dependency on successful clustering, and a shortcoming may occur when clustering algorithms struggle due to data complexity.

Moving forward, we aim to focus on developing enhanced algorithms with an emphasis on more evenly distributing the dataset. By improving the partitioning of the data, we hope to increase the robustness and applicability of our approach in diverse data scenarios. In doing so, we anticipate further optimizing our method's capacity to accurately discern causal structures, thereby advancing the field's understanding and modeling of complex systems.

## 7 Acknowledgement

This work was in part supported by the National Science Foundation (award 2318101), C-BRIC (one of six centers in JUMP, a Semiconductor Research Corporation (SRC) program sponsored by DARPA) and the Army Research Office (W911NF2020053). The authors affirm that the views expressed herein are solely their own, and do not represent the views of the United States government or any agency thereof.

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

## Appendix

## A    Selection of ISL hyperparameters

**Thresholds:** As described in Sec. 3.2 and Algorithm 1, after Eq.3 converges at all environments, we employ a threshold $t$ to convert the adjacency matrix $W$ to a DAG. To find a proper threshold, we use the following strategy. We set a minimum edges number $E_{min}$ and a maximum edges number $E_{max}$ based on the dataset information. Usually, $E_{min}$ is half of the number of nodes $|E|/2$ and $E_{max}$ is $5|E|$. We also set a range of threshold $t \in [t_{min}, t_{max}]$ and a step size $t_s$ base on the value range of $W$. Usually we use $t_{min} = \min(W)$ and $t_{max} = \max(W)$. Then, we employ a grid search over the range $[t_{min}, t_{max}]$ with a step size $t_s$, and keep the thresholds and corresponding DAG that satisfied the following requirements: (1) The graph after the filtering with threshold should be a DAG (no cyclicity). (2) The number of graph edges $E_{min} < E < E_{max}]$. For the selected threshold values and DAGs, we remove the duplicated group as different threshold may obtain the same DAG, which further refines the interval. Then, for each threshold, we use the selected $Pa(Y)$ as input to train a one-layer MLP to predict $Y$ and select the threshold $t$ that has smallest $Y$ reconstruction error in the validation set.

**Regularization coefficients:** For training of ISL, we use different loss terms. The hyperparameter $\gamma$ controls the trade off between $Y$ reconstruction and DAG constrain among environments. As we decrease the value of $\gamma$, the training would focus more on the target $Y$ reconstruction. We also have 4 regularization hyperparameters: $\mathcal{L}_{sparse}(\theta) = \beta_1 ||\theta_1^Y||_1 + \beta_2 ||\theta_r^Y||_2 + \beta_3 ||\theta^X||_1 + \beta_4 ||\theta^X||_2$, where $||\cdot||_1$ and $||\cdot||_2$ denote $l_1$ and $l_2$ regularization. $\beta_1$ controls the importance of the $l_1$ regularization on the $\theta_1^Y$, increasing $\beta_1$ makes the selection of $Pa(Y)$ more conservative (most of the values of the first column in $W$ would be zero). $\beta_2$ helps avoid overfitting of $h(\cdot)$. $\beta_3$ and $\beta_4$ controls the regularization on $\theta^X$. We choose the value of $\gamma$ and $\beta_i$ that achieves the smallest target Y reconstruction on the validation set. We find the parameters: $\gamma = 1; \beta_1 = 0.001; \beta_2 = 0.01; \beta_3 = 0.01; \beta_4 = 0.01$ as reasonable choices across many different settings, although they are not extensively optimized.

Table. 6 shows the results on Boston Housing for the prediction target of median value of homes (MED) ISL with different regression parameters. We demonstrate that the results are not too sensitive to the change of regularization. That is because the regularization coefficients mainly influence the DAG learning process, and we apply fine-tuning for $h(\cdot)$ after convergence of DAG learning, which provides a mechanism to mitigate the differences at the first training stage.

Table 6: Boston Housing median value of homes (MED) target prediction results by ISL with different regression parameters.

| $\gamma$ | $\beta_1$ | $\beta_2$ | $\beta_3$ | $\beta_4$ | MSE ($\downarrow$) |
|---|---|---|---|---|---|
| 1 | 0.001 | 0.01 | 0.01 | 0.01 | 0.052 |
| 5 | 0.001 | 0.01 | 0.01 | 0.01 | 0.054 |
| 1 | 0.001 | 0.001 | 0.01 | 0.01 | 0.057 |

## B    Building environments

To show the efficacy of the proposed unsupervised environment building method based on k-means clustering, we present comparisons to the setting with the environment building based on known data source information, i.e. the data comes with the indication on how the environments are split based on data collection or generation process. Table. 7 shows that their difference is quite small (much smaller than the outperformance of ISL compared to the other methods) and the proposed ISL is highly effective and robust with unsupervised environment building by clustering.

We employed standard clustering evaluation techniques such as the Elbow and Silhouette Method. These methods enabled us to assess various cluster numbers systematically, identifying the point at which adding

more clusters led to diminishing returns (the "elbow") and evaluating how well each object lies within its cluster (the silhouette score). More specifically, for the Elbow method, we plot the sum of squared distances from each point to its assigned center (known as inertia or within-cluster sum of squares) for a range of values of K (e.g., K from 1 to 10) and automatically detect the "elbow" point on the plot, where the reduction in inertia starts to slow down. For the silhouette method, we calculate its silhouette score, which measures how similar a point is to its cluster compared to other clusters for each data point, and compute the average silhouette score for different values of K and plot them. An algorithm is then applied to look for an optimal value K that maximizes the average silhouette score across all data points. This process allowed us to arrive at an optimal K value, and our extensive experimentation and analysis demonstrated that our approach remains robust across different numbers of environments. Our empirical evaluations provide further evidence of the effectiveness and stability of our selection method.

Table 7: Unsupervised vs. supervised environment building for ISL on synthetic data. We observe very small difference between them, showing the efficacy of the proposed unsupervised environment construction mechanism.

| Number of nodes | Metrics ($\downarrow$) | Supervised | Unsupervised |
|---|---|---|---|
| 3 (c=2, s=1) | ID MSE | **0.005** ±0.0001 | **0.005** ±0.0001 |
| | OOD MSE | **0.010** ±0.0002 | **0.010** ±0.0002 |
| | Average SHD | **0**±0 | **0**±0 |
| 4 (c=2, s=2) | ID MSE | **0.006** ±0.0002 | **0.006** ±0.0002 |
| | OOD MSE | **0.009** ±0.0001 | **0.009** ±0.0001 |
| | Average SHD | **0**±0 | **0**±0 |
| 5 (c=3, s=2) | ID MSE | **0.004** ±0.0001 | **0.004** ±0.0001 |
| | OOD MSE | **0.004** ±0.0001 | **0.004** ±0.0001 |
| | Average SHD | **0**±0 | **0**±0 |
| 9 (c=4, s=5) | ID MSE | **0.004** ±0.0006 | **0.004** ±0.0006 |
| | OOD MSE | **0.005** ±0.0001 | **0.005** ±0.0001 |
| | Average SHD | **0**±0 | **0**±0 |
| 20 (c=10, s=10) | ID MSE | **0.007** ±0.0005 | **0.009** ±0.0005 |
| | OOD MSE | **0.007** ±0.0001 | **0.061** ±0.009 |
| | Average SHD | **1**±0 | **2**±1 |

## C   Error statistics

In this section, we present the standard deviations for the errors to highlight the statistical significance of ISL improvements (Table. 8 and Table. 9). Overall, the improvements of ISL are much larger than the standard deviation values. In addition, the variance of performance is observed to be lower for ISL compared to the other methods, indicating its superiority in robustness.

Table 8: Supervised learning experiment results on real-world data along with their standard deviations. Note that MLP and CASTLE cannot provide DAGs (and thus don't have SHD values).

| Dataset | Target | MSE ($\downarrow$) | | | | SHD ($\downarrow$) | |
|---|---|---|---|---|---|---|---|
| | | MLP | NOTEARS-MLP | CASTLE | ISL (Ours) | NOTEARS-MLP | ISL (Ours) |
| Boston Housing | MED | 0.16 ±0.02 | 0.12 ±0.03 | 0.10 ±0.01 | **0.05** ±0.008 | 2 ±0 | **1**±0 |
| Insurance | 'PropCost' | 0.40 ±0.02 | 0.99 ±0.02 | 0.36 ±0.001 | **0.34** ±0.004 | 2 ±0 | **0**±0 |
| | 'MedCost' | 0.69 ±0.09 | 1.03 ±0.01 | 0.55 ±0.03 | **0.52** ±0.002 | 2 ±1 | **0**±0 |
| | 'LiabilityCost' | 0.94 ±0.08 | 0.39 ±0.01 | 0.38 ±0.06 | **0.25** ±0.0004 | 1 ±0 | **0**±0 |
| | 'CarValue' | 0.23 ±0.01 | 0.60 ±0.05 | 0.23 ±0.03 | **0.23** ±0.0004 | 2 ±0 | **1**±0 |

Table 9: Synthetic tabular data experiments in supervised learning setting. Note that black-box MLP and CASTLE can't provide DAGs. ISL yields lower MSE for ID and OOD, and lower SHD.

| Number of nodes | Metrics (↓) | MLP | NOTEARS-MLP | CASTLE | ISL (Ours) |
|---|---|---|---|---|---|
| 3 (c=2, s=1) | ID MSE | 0.008 ±0.002 | 0.101 ±0.010 | 0.016 ±0.007 | **0.005** ±0.0001 |
| | OOD MSE | 0.016 ±0.002 | 0.195 ±0.005 | 0.017 ±0.004 | **0.010** ±0.0002 |
| | Average SHD | - | 2 ±0 | - | **0**±0 |
| 4 (c=2, s=2) | ID MSE | 0.006 ±0.009 | 0.087 ±0.005 | 0.017 ±0.002 | **0.006** ±0.0002 |
| | OOD MSE | 0.040 ±0.022 | 0.174 ±0.024 | 0.036 ±0.010 | **0.009** ±0.0001 |
| | Average SHD | - | 2 ±0 | - | **0**±0 |
| 5 (c=3, s=2) | ID MSE | 0.004 ±0.002 | 0.110 ±0.018 | 0.025 ±0.006 | **0.004** ±0.0001 |
| | OOD MSE | 0.004 ±0.002 | 0.078 ±0.020 | 0.019 ±0.004 | **0.004** ±0.0001 |
| | Average SHD | - | 3 ±0 | - | **0**±0 |
| 9 (c=4, s=5) | ID MSE | 0.012 ±0.006 | 0.070 ±0.010 | 0.034 ±0.010 | **0.004** ±0.0006 |
| | OOD MSE | 0.052 ±0.024 | 0.201 ±0.028 | 0.152 ±0.022 | **0.005** ±0.0001 |
| | Average SHD | - | 4 ±0 | - | **0**±0 |
| 20 (c=10, s=10) | ID MSE | 0.009 ±0.008 | 0.061 ±0.011 | 0.121 ±0.021 | **0.007** ±0.0005 |
| | OOD MSE | 0.094 ±0.061 | 0.303 ±0.050 | 0.272 ±0.046 | **0.007** ±0.0001 |
| | Average SHD | - | 9 ±0 | - | **1**±0 |

## D Time Complexity and Scalability Comparison

With respect to runtime and scalability, our runtime is 3 to 5 times greater than that of NOTEAR **?** due to the additional clustering step. The computational complexity of our framework aligns closely with that of NOTEARS. Specifically, the complexity for NOTEARS-MLP is given by $O(nd^2m + d^2m + d^3)$ FLOPS per iteration of L-BFGS-B, where $n$ is the number of data samples, $d$ is the number of nodes, and $m$ is the number of edges.

In our framework, as we partition the dataset into $k$ different environments, each requiring its own convergence, the effective runtime becomes $O(k(nd^2m + d^2m + d^3))$. However, since $k$ (the number of environments) is generally a small constant—typically ranging from 3 to 5—the overall time complexity remains on the same order as $O(nd^2m + d^2m + d^3)$.

The following table summarizes some of the quantitative results we have recorded. The time measurements were obtained on an Apple M1 Pro chip with 16GB of memory.

| Experiment | NOTEAR time (s) | ISL (ours) | Self-supervised ISL (ours) |
|---|---|---|---|
| x2s1 | 32.6 | 100.5 | 280.5 |
| x2s2 | 48.1 | 139.7 | 400.2 |
| x3s2 | 62.9 | 188.5 | 520.8 |
| x4s5 | 139.0 | 420.1 | 1220.3 |
| x10s10 | 414.5 | 1300.4 | 3600.4 |

Table 10: Time benchmarks for NOTEAR and our proposed methods.

## E Discussion

While we have indeed demonstrated our idea through an image classification task, this example serves more to elucidate the intuition of our algorithm rather than showcase a specific target application. Our work primarily focuses on tabular data because most of the real world domains like economics, biology, and social social that we are interested in applying the causal discovery algorithm gather data in structured, tabular form and most causal discovery algorithms such as conditional independence tests and structural learning algorithms are designed to work with structured data. We have evaluated our approach on two well-known real-world

datasets: the Sachs dataset [9] and the Insurance dataset. Both of these datasets are widely recognized and extensively used as benchmarks in the field of causal discovery (Zheng et al., 2018; 2020; He et al., 2021; Wei et al., 2020; Yu et al., 2019). We have clarified this in the revised manuscript.

## F    Limitations and the societal impact

In this paper, we propose a novel method for causal structure discovery, which can improve the explainability and generalization of key machine learning use cases. Lack of their explainability remains to be a bottleneck for widespread adoption of DNNs for many high-stakes applications, such as from Healthcare, Finance, Public Sector, Insurance, Legal etc. There are other forms of explainability methods used in practice, but since they cannot explicitly distinguish the causality from the correlations, there are many cases that they cannot satisfy the high bar for explainability in such applications. We believe that our method constitutes an important contribution towards this, as it can be directly adopted to applications where obtaining accurate causal explanations is crucial. In some cases, causal explanations can uncover the undesired biases in the data such as when the dominant factor for the output label comes from one of the features that corresponds to a sensitive attribute such as gender. In these cases, the causal explanations can be further validated with additional analyses (as our model is still far away from achieving the perfect SHD of 0 on complex real-world data with many features), and if they seem to be convincing, further data manipulation or model debiasing actions can be performed. In addition to causal explainability, the improved generalization aspect is expected to play a major positive role, as the distribution differences between training and testing settings can sometimes hinder the reliability of machine learning models. In some applications where the data collection is limited to certain locations or times or subsets, our method can be utilized to enhance the performance of the trained models when they are deployed to operate for different locations or times or subsets.

Overall, we believe there is significant room for improvement in causal structure discovery. Especially for complex real-world data with many features, the obtained SHD values are not very low in the literature. Further research in unsupervised environment building with better representation learning, end-to-end approaches in combining graph discovery and supervised learning, and adding more nonlinearity to the model to make it higher capacity in a systematic way, can be promising towards this direction. We demonstrate the robustness of our model in various settings, but further exploration of theoretical convergence guarantee can be useful as well. Lastly, methods to improve hyperparameter tuning and model selection with small validation data, without relying on ground truth causal graph structure, would be of high value.

