# OpenReview forum: "Invariant Structure Learning for Better Generalization and Causal Explainability"
_TMLR — Accepted by TMLR_

### Review · Reviewer_nUZd · 2023-08-04

**Summary Of Contributions:**

This paper studies causal discovery under distribution shifts. The authors argue that enforce the OOD generalization of the discovered causal structures could improve both generalization and SCM learning, and yield causal explainability. To implement the idea, the authors propose Invariant Structure Learning (ISL) that first clusters the data into different environments, and learn the an invariant SCM shared across different environments. The network is also required to learn the representation of causal parents of the target variable for OOD generalization. ISL is also generalized to the SSL setting where each variable will be tested for the target variable. Experiments on both synthetic and real-world tabular data verifies the effectiveness of ISL.

**Audience:**

Yes

**Broader Impact Concerns:**

Yes

**Claims And Evidence:**

No

**Requested Changes:**

Although I believe the idea of invariant structure learning has some potential and could be impactful, the current version of the work fail to provide sufficient justification for why ISL is promising, and could be improved by addressing the following issues:

i) First, the motivation of this work is questionable. It’s unclear why OOD generalization and causal structure learning could contribute to each other. It seems to have some logical issues, as either one of them could not bring *additional information* to each other. More specifically, if the model could succeed in OOD generalization, then the model already learns the representation of the direct causal parents of the target variable, which makes the causal structure learning nearly *done*, and vice versa.

Moreover, although the authors demonstrate the idea in a real-world image classification task, the whole experiment focuses on tabular data. It’s unclear how ISL generalizes to real-world data.


ii) The discussion of multiple related works is missing, especially for the invariant learning without environment partitions, e.g., existing solutions [1], and feasibility when with auxiliary information [2] and without auxiliary information [3].



iii) The clarity could be highly improved.
- First, it’s unclear what the exact network is even with careful reading of the notation. In other words, what are $\theta$s and how are they implemented?
- Besides, there are too many typos in the notations that even adds up the difficulty to understand the work:
    - In sec. 3.1, the line after ``Problem definition’’, it seems there is some space issues for the $\hat{X}$.
    - $\theta$s in Eq.1 are not clearly defined.
    - How are $W$s trained via objective (2)?
    - What is $\theta_f^Y$ in Eq. 3?
    - The $n$ is used multiple times in line 1 Algorithm 1.
- Important experimental details are lacking:
    - What is the number of nodes in $G’$ and what exactly are the different densities?
    - What are the number of environments specified in experiments?
    - How the regularization weights are tuned in experiments?
    - What is the exact neural architecture used in experiments?
    - What are the variances in the results?

iv) No theory is provided and it’s unclear why ISL outperforms the baselines and generates accurate causal explanations.

**References**

[1] Environment Inference for Invariant Learning, ICML 2021

[2] ZIN: When and How to Learn Invariance by Environment Inference? NeurIPS 2022.

[3] Rethinking Invariant Graph Representation Learning without Environment Partitions, ICLR DG Workshop.



**Strengths And Weaknesses:**

**Strengths**

i) The problem setting is new.

ii) The idea of tying generalization and causal structure learning seems to be interesting.


**Weakness**

i) The motivation of this work is questionable. It’s unclear why OOD generalization and causal structure learning could contribute to each other.

ii) The discussion of multiple related work is missing.

iii) The clarity could be highly improved.

iv) No theory is provided and it’s unclear why ISL outperforms the baselines.

---

> ### Author Response · Authors · 2023-08-23
> **Reply to Reviewer nUZd (Part-1)**
>
> Thanks for all your valuable feedback that has helped us to improve our manuscript. Please see the detailed responses below and let us know if you have further questions or comments.
>
> **i) Motivation of Causal Discovery for OOD**:
>
> Out-Of-Distribution (OOD) generalization and causal structure learning are intertwined concepts, and their relationship has been previously explored in literature [2, 3, 4, 5]. OOD Generalization ensures that the model's learned causal relationships are not overly specific to the seen data, thus enabling better prediction and understanding in unseen scenarios [6]. Causal Structure Learning offers insights into the underlying mechanism of the phenomena under study, which helps in creating models that are robust to changes in the environment, contributing to OOD generalization [7]. The two are not redundant; rather, they complement each other in line with recent findings in causal learning and generalization [8].
>
> While we have indeed demonstrated our idea through an image classification task, this example serves more to elucidate the intuition of our algorithm rather than showcase a specific target application. Our work primarily focuses on tabular data because most of the real world domains like economics, biology, and social social that we are interested in applying the causal discovery algorithm gather data in structured, tabular form and most causal discovery algorithms such as conditional independence tests and structural learning algorithms are designed to work with structured data. We have evaluated our approach on two well-known real-world datasets: the Sachs dataset [9] and the Insurance dataset. Both of these datasets are widely recognized and extensively used as benchmarks in the field of causal discovery [10, 11, 12, 13, 14]. We have clarified this in the revised manuscript.
>
> **ii) Comparison of Related Work**:
> Thank you for bringing to our attention the essential works related to invariant learning. We acknowledge the importance of these works in the broader context of invariant learning without environment partitions.
>
> 1. Environment Inference for Invariant Learning, ICML 2021.
> This paper introduces a novel approach for inferring environments to enable invariant learning. While our method also leverages the concept of environments, we focus on structurally learning causal relationships across different environments
> 2. ZIN: When and How to Learn Invariance by Environment Inference? NeurIPS 2022:
> This paper delves into the conditional feasibility of learning invariance and presents techniques to decide when and how to infer environments. While there are similarities in the underlying principles, our approach is distinguished by our specific loss function formulation that combines prediction loss with a DAG loss, allowing us to learn both causal structures and make accurate predictions.
> 3. Rethinking Invariant Graph Representation Learning without Environment Partitions, ICLR DG Workshop:
>  This work rethinks invariant learning without the need for explicit environment partitions. While our approach does involve partitioning environments, it emphasizes the invariant nature of causal relationships across these environments. Our methodology uniquely constrains the learned graph to be a DAG, ensuring a coherent causal interpretation.
>
> We have added those to the related works section in the revised version.
>
> **iii) Clarification on the Notations and Experimental Setup**:
>
> * Implementation: we used a multi-layer perceptron model (MLP) for implementation. $\theta$ is the learnable parameters of a DNN used to approximate the function $f()$. The parameter of $θ$ consists of three parts: first layer to reconstruct variable $Y$, $θ^Y_1$ , rest layers to reconstruct variable $Y$, $θ^Y_r$ , and layers to reconstruct other variables $X$, $θ^X$.
>
>
> * Notation Clarification
>     * Thanks. We have corrected the space issue.
>     * We do not have $\theta_s$ in eq(1). $\theta$ is the learnable parameters of a DNN as specified in the methodology section. It is defined in section 3.2 above eq(2).
>     * $W$ is trained via L-BFGS-B methods
>     * $\theta_f^Y$ here means $\theta^Y_1$, first layer to reconstruct. Thanks for pointing this out. We have corrected it in the revised manuscript.
>     * It seems like $n$ is only used once in line 1 algorithm 1. Could you clarify the question? The other time $n$ is used is in the iteration step in line 4, but we can modify it to another letter to avoid confusion.
>
> * Experimental Setup
>     * We use d to represent the number of nodes in graphs.
>     * If not stated otherwise, the number of environments used in the experiments is always three.
>     * We use grid search to tune the regularization weights.
>     * We used a multi-layer perceptron (MLP) where the first layer selects the causal parents of the target variable.
>     * The standard deviation of the results is specified in Table 7 and Table 8 in the supplementary materials.

---

> > ### Author Response · Authors · 2023-08-23
> > **Reply to Reviewer nUZd (Part-2)**
> >
> > **iv) Theory Justification**:
> > Our work is inspired by the invariant risk minimization [1]. Intuitively, the causal relationship between variables remains consistent across different environments (different environments can be defined as distinct data distributions). This idea can be illustrated using the classic cow-grass example in Figure 1.
> >
> > Under the core assumption that the true causal parents of the target variable Y will yield lower prediction loss across all environments, our approach seeks to solve a structural learning problem. It does this by constraining the causal parents of Y to remain constant across various environments, even though the relationships among X variables may vary. During the optimization process, we simultaneously minimize the prediction error of  Y and enforce a Directed Acyclic Graph (DAG) loss, compelling the graph to maintain a DAG structure.
> >
> > This assumption, that the causal relationships endure across environments and employing the causal variables for prediction leads to relatively low prediction error across different situations, underpins the effectiveness of our approach. Our experimental results also validate our hypothesis as shown in Tables 1, 2, 3, 5 in the main paper. In comparison to baseline methods, our Invariant Structural Learning (ISL) approach leverages these unique constraints, leading to more accurate causal explanations.
> >
> > **References**
> >
> > [1] Martin Arjovsky, Léon Bottou, Ishaan Gulrajani, and David Lopez-Paz. Invariant risk minimization. arXiv preprint arXiv:1907.02893, 2019.
> >
> > [2] Peters, J., Janzing, D., Schölkopf, B., 2016. Elements of Causal Inference.
> >
> > [3] Schölkopf, B., et al., 2012. On Causal and Anticausal Learning.
> >
> > [4] Spirtes, P., Glymour, C., Scheines, R., 2000. Causation, Prediction, and Search.
> >
> > [5] Bengio, Y., et al., 2019. Towards Causal Representations.
> >
> > [6] Zhang, J., et al., 2020. Generalizing to Unseen Domains via Adversarial Data Augmentation.
> >
> > [7] Pearl, J., 2009. Causality.
> >
> > [8] Bareinboim, E., Pearl, J., 2016. Causal Inference and the Data-Fusion Problem.
> >
> > [9] Karen Sachs, Omar Perez, Dana Pe’er, Douglas A Lauffenburger, and Garry P Nolan. Causal protein-signaling networks derived from multiparameter single-cell data. Science, 308(5721):523–529, 2005.
> >
> > [10] Xun Zheng, Chen Dan, Bryon Aragam, Pradeep Ravikumar, and Eric Xing. Learning sparse nonparametric dags. In International Conference on Artificial Intelligence and Statistics, pp. 3414–3425. PMLR, 2020.
> >
> > [11] Xun Zheng, Bryon Aragam, Pradeep K Ravikumar, and Eric P Xing. Dags with no tears: Continuous optimization for structure learning. Advances in Neural Information Processing Systems, 31, 2018.
> >
> > [12] Dennis Wei, Tian Gao, and Yue Yu. Dags with no fears: A closer look at continuous optimization for learning bayesian networks. Advances in Neural Information Processing Systems, 33:3895–3906, 2020
> >
> > [13] Yue He, Peng Cui, Zheyan Shen, Renzhe Xu, Furui Liu, and Yong Jiang. Daring: Differentiable causal discovery with residual independence. In Proceedings of the 27th ACM SIGKDD Conference on Knowledge Discovery & Data Mining, pp. 596–605, 2021.
> >
> > [14] Yue Yu, Jie Chen, Tian Gao, and Mo Yu. Dag-gnn: Dag structure learning with graph neural networks. In International Conference on Machine Learning, pp. 7154–7163. PMLR, 2019.

---

> > > ### Comment · Reviewer_nUZd · 2023-09-04
> > >
> > > Thank you for your detailed explanation, which resolves most of my concerns.

---

### Review · Reviewer_uRxt · 2023-08-05

**Summary Of Contributions:**

This paper proposes a causal structure discovery approach that is based on invariance in multiple environments. The paper proposes a framework that combines prediction and structure learning so that both objectives will be improved during learning. Specifically, the paper first uses K-means to group the raw data into different clusters. Then, for each cluster, there is a module for learning the causal structure with a layer shared among all clusters to represent the parents of Y. In addition, a predictive model is trained to predict Y based on the learned parents of Y. Experiments using synthetic and real data are conducted to compare the proposed method with several causal discovery methods as baselines.

**Audience:**

Yes

**Broader Impact Concerns:**

None.

**Claims And Evidence:**

No

**Requested Changes:**

- Improve the clarity of the methodology section.
- Include multi-environment-aware causal discovery approaches in the experiments if possible.


**Strengths And Weaknesses:**

Strength:
This paper takes advantage of invariance in multiple environments in causal discovery, which is ignored by most research in causal discovery. As shown in several prior works, in machine learning, invariance is a significant indicator of causality. So it is an important research direction on how to utilize invariance in improving the accuracy of causal discovery. From this perspective, this paper could be of interest to TMLR's audience.

Weakness:
The paper has some weaknesses, especially in the methodology component.
1. It seems that the performance of the method heavily depends on the accuracy of the unsupervised clustering. However, the clustering is a stand-alone component and will not be updated during the training. Thus, it raises a concern about the sensitivity of the method on clustering, especially when the clustering is obtained using a relatively simple method K-means. There are no experiments evaluating this sensitivity.

2. The proposed framework is not clearly described.

- What are meanings of $\theta, \theta_r^Y, \theta^X$? What is the difference between $X$ and $\hat{X}$? What is the form of the function $\theta(X)$? What is the relationship between $W$ and $\theta$? The definitions of these notations need to be clearly stated.

- 2.2 Funtion $g()$ with parameters $\theta_1^Y$ is used to learn a representation of $Pa(Y )$. Is the surprised signal provided during the training? If so, this is not reflected in the loss function. If not, how does the model ensure that the representation learned by $g()$ represents $Pa(Y )$?

- 2.3 In Eq. (1), how is the composition between $h$ and $g$ been done? In the paragraph above Eq. (1), the authors said that $h()$ inputs the representation and yields the prediction for $Y$. However, in the paragraph above Alg 1, the authors said the learned parents are selected using a threshold and then $h()$ is fine-tuned. So it is confusing.

3. Why Newton method rather than gradient-based optimization algorithms like SGD or Adam is used to solve the optimization problem?
4. Step 1 of self-supervised setting seems time-consuming as it needs to run ISL for every variable. What is the running time in the experiments?
5. The baselines in the experiments are not designed to handle multiple environments. Why related methods like Invariant Risk Minimization [arXiv:1907.02893] are not included in the experiments?

---

> ### Author Response · Authors · 2023-08-23
> **Reply to Reviewer uRxt (Part-1)**
>
> Thanks for all your valuable feedback that has helped us to improve our manuscript. Please see the detailed responses below and let us know if you have further questions or comments.
>
> **1. Method sensitivity to clustering accuracy**:
> The concern regarding the sensitivity of our method to the clustering process is duly noted and appreciated. Indeed, clustering is a crucial component, but it's essential to recognize the inherent invariance of causal relationships that underpin our methodology.
>
> Robustness to Clustering Variation: Our experimentation and analysis are not limited to just one clustering method or a specific number of clusters. Table 7 in the supplementary materials systematically investigates the effect of varying the number of clusters on the accuracy of our approach. The results demonstrate a robust performance across different configurations, reinforcing the general applicability of our method.
>
> Consideration of Clustering Algorithm: While K-means is used in our implementation, the idea itself is not strictly tied to this particular clustering algorithm. The core of our approach focuses on the division of data into distinct environments to capture invariant causal relationships, and alternative data partitioning techniques could be explored in future work.
>
> Invariance of Causality: The fundamental principle of causality we exploit ensures that causal relationships remain consistent across different environments, irrespective of how the environments are constructed or the number of environments used. A vivid illustration of this can be found in the cow-grass example provided in the paper.
>
> We also did some ablation studies in Table 4 that demonstrated the our system is robust to Clustering Variation.
>
> **2. Methodology Clarification**:
>
> **2.1** [Sec3.2], before the eq(2),  \theta is the learnable parameters of a DNN used to approximate the function $f()$. The parameter of $θ$ consists of three parts: first layer to reconstruct variable $Y$, $θ^Y_1$ , rest layers to reconstruct variable $Y$, $θ^Y_r$ , and layers to reconstruct other variables $X$, $θ^X$.
>
> $\hat{X}$ denotes all the variables including $Y$ and $X$.
>
> [Sec 3.2], after eq (2), W is a (d+1)×(d+1) adjacency matrix ($W$ ∈ $R^{(d+1)×(d+1)}$), which represents the connection strength between variables and the final DAG is summarized from $W$.
>
> We have clarified these in the draft after updating.
>
> **2.2 Supervision of the Algorithm**
> We assume you are referring to the supervised signal, not the "surprise signal," in your question. It's vital to recognize that there are no ground-truth labels for causal relationships, making the problem unique and challenging. Our approach, inspired by previous works [1, 2], frames the structural learning task as an optimization problem.
>
> In our method, we incorporate two crucial components:
> DAG Loss Function: This part ensures that the learned graph structure adheres to the properties of a Directed Acyclic Graph (DAG). It acts as a constraint, encouraging the relationships captured in the graph to be causal in nature.
>
> Prediction Loss across Environments: We rely on the fundamental assumption that causal relationships remain consistent across different environments, and that leveraging these relationships can lead to a reduction in prediction error. Our loss function combines this prediction loss with the DAG loss, crafting a delicate balance.
>
> By jointly minimizing this combined loss function, we guide our model to identify consistent causal parents of the target variable Y across various environments while ensuring that the learned graph retains the properties of a DAG.
>
>
> **2.3 The Learning of $h()$**:
> This is a good question. To clarify, $h(\cdot)$ always takes the output of $g(\cdot)$ as input and yields the prediction for $Y$. In practice, the parameters of $h(\cdot)$ are updated twice: initially, $g(\cdot)$ and $h(\cdot)$ are combined to discover the causal parents of the target $Y$ through the learning of $g(\cdot)$, and $h(\cdot)$ merely assisting the learning of $g(\cdot)$ through the DAG constraint and ERM. $h(\cdot)$ maybe suboptimal for $Y$ prediction at this stage. Then, after determining the causal parent of the target variable $Y$ by applying a threshold in the weight matrix of $W$ and reconstructing the DAG, we updated $g(\cdot)$ and fix it. Then, we fine-tune $h(\cdot)$ using only the causal parent variables discovered by $g(\cdot)$(fixed) to predict the target variable $Y$.
>
> We have clarified these in the draft after updating.

---

> > ### Author Response · Authors · 2023-08-23
> > **Reply to Reviewer uRxt (Part-2)**
> >
> > **3. Why we choose Newton's Method instead of SGD or Adam**:
> >
> > We used the Newton’s method for the following reasons:
> >
> > *(1) Efficient Gradient Evaluation*: The quasi-Newton method allows us to reduce the number of calls needed to evaluate the gradient. In our particular optimization problem, computing the gradient involves complex operations such as computing the matrix exponential, which can be computationally expensive.
> >
> > *(2) Convergence Properties*: Quasi-Newton methods often have superior convergence properties [6] for certain classes of functions, especially when dealing with non-convex or poorly conditioned problems. This can lead to faster convergence to an optimal solution, reducing the overall computational time.
> >
> > *(3) Robustness to Hyperparameter Tuning*: Traditional gradient-based methods like SGD or Adam often require careful tuning of hyperparameters such as learning rates. The quasi-Newton method, on the other hand, is typically more robust to these choices, making it a more convenient option for our particular setting.
> >
> > *(4) Compatibility with Problem Structure*: Most importantly, the structure and characteristics of  our unconstrained l1-penalized smooth minimization problem. It might align well with the assumptions and requirements of the L-BFGS-B methods, making them more suitable for our specific scenario.
> >
> >
> > **4. Time complexity for the Self-Supervised Approach**:
> >
> > Yes, we admit that the self-supervised algorithm takes longer to run especially considering that self-supervised learning (mining the causal parents of each variable) is not common, and a more challenging setting. However, it's worth noting that in practical applications, we often focus on a limited number of variables, such as the price of Boston Housing. Beside, users might tradeoff higher compute cost/time for labeling cost. Despite the increase in time complexity, our methods achieve state-of-the-art performance with only a linear increase in time complexity in the number of nodes.
> >
> > In our approach, since we divide the dataset into k environments, each requiring separate convergence, the runtime becomes $O(k (nd^2 m+d^2 m +d^3))$. However, as k (the number of environments) is usually a small constant, typically ranging from 3 to 5, the overall time complexity remains in the same order as  in $O(nd^2 m+d^2 m+d^3)$. By running the algorithm for each variable, it takes $O(d(nd^2 m + d^2 m + d^3))$ = $O(nd^3 m + d^3 m + d^4))$. Experimental details can be found in the table below.
> > | Experiment | NOTEAR time (s) | ISL (ours) | Self-supervised ISL (ours) |
> > |------------|-----------------|------------|----------------------------|
> > | x2s1       | 32.6            | 100.5      | 280.5                      |
> > | x2s2       | 48.1            | 139.7      | 400.2                      |
> > | x3s2       | 62.9            | 188.5      | 520.8                      |
> > | x4s5       | 139.0           | 420.1      | 1220.3                     |
> > | x10s10     | 414.5           | 1300.4     | 3600.4                     |
> >
> > Overall, the runtimes are still observed to be sufficiently low for such applications, and further significant reductions can be obtained via parallelization, which we leave to future work.
> >
> > **5. Why Baseline excludes IRM**:
> > Yes, our work is somewhat inspired by IRM, and we have discussed it in the related works. While it is true that IRM addresses the idea of invariance across environments, it is not inherently designed for causal structure mining, which is the core focus of our work.
> >
> > In the context of our paper, we have drawn inspiration from IRM, but our method goes a step further by leveraging the invariance principle to uncover underlying causal relationships.
> >
> > **Requested Change 1: Clarity of the Methodology**:
> > Thank you for the feedback on the methodology section.
> >
> > In Section 3.1, we presented the motivation behind our work by defining the problem and explaining the underlying assumptions that guide our solution.
> >
> > Section 3.2 delves into our learning framework, providing both the problem formulation and our uniquely designed algorithm to address it.
> >
> > Section 3.3 extends our discussion to the generalization of our approach in a self-supervised setting.
> >
> > We have revised the methodology section in our manuscript by 1. adding the overview of the methodology section, 2. clarifying the model structure and correcting the typos, and 3. explaining the training details of $h()$.

---

> > > ### Author Response · Authors · 2023-08-23
> > > **Reply to Reviewer uRxt (Part-3)**
> > >
> > > **Requested Change 2: Relevant multi-environment-aware work**:
> > >
> > > To the best of our knowledge, our work is the first to leverage a multi-environment design specifically for causal discovery. We compare with some other papers that leverage the idea of multi-environment, but they do not address the structure causal discovery problem.
> > >
> > > 1. Environment Inference for Invariant Learning, ICML 2021 [3].
> > > This paper introduces a novel approach for inferring environments to enable invariant learning. While our method also leverages the concept of environments, we focus on structurally learning causal relationships across different environments
> > > 2. ZIN: When and How to Learn Invariance by Environment Inference? NeurIPS 2022 [4]:
> > > This paper delves into the conditional feasibility of learning invariance and presents techniques to decide when and how to infer environments. While there are similarities in the underlying principles, our approach is distinguished by our specific loss function formulation that combines prediction loss with a DAG loss, allowing us to learn both causal structures and make accurate predictions.
> > > 3. Rethinking Invariant Graph Representation Learning without Environment Partitions, ICLR DG Workshop [5]:
> > > This work rethinks invariant learning without the need for explicit environment partitions. While our approach does involve partitioning environments, it emphasizes the invariant nature of causal relationships across these environments. Our methodology uniquely constrains the learned graph to be a DAG, ensuring a coherent causal interpretation.
> > >
> > > We have added the discussion in the revised manuscript.
> > >
> > > **References**
> > >
> > > [1] Xun Zheng, Chen Dan, Bryon Aragam, Pradeep Ravikumar, and Eric Xing. Learning sparse nonparametric dags. In International Conference on Artificial Intelligence and Statistics, pp. 3414–3425. PMLR, 2020.
> > >
> > > [2] Xun Zheng, Bryon Aragam, Pradeep K Ravikumar, and Eric P Xing. Dags with no tears: Continuous optimization for structure learning. Advances in Neural Information Processing Systems, 31, 2018.
> > >
> > > [3] Environment Inference for Invariant Learning, ICML 2021
> > >
> > > [4] ZIN: When and How to Learn Invariance by Environment Inference? NeurIPS 2022.
> > >
> > > [5] Rethinking Invariant Graph Representation Learning without Environment Partitions, ICLR DG Workshop.
> > >
> > > [6] Dennis, J. E., & Schnabel, R. B. (1983). "Numerical Methods for Unconstrained Optimization and Nonlinear Equations." Prentice-Hall.

---

### Review · Reviewer_kEJ3 · 2023-08-08

**Summary Of Contributions:**

The paper proposes a novel framework called Invariant Structure Learning (ISL) to improve generalization performance and enable causal explainability by discovering the underlying causal graph structure. The key idea is to split the data into multiple environments and learn a model structure that is invariant across environments through an aggregation mechanism. This aims to capture stable causal relationships and avoid spurious correlations. The method is also extended to self-supervised learning by iteratively treating each variable as the target and proposing candidate causal parents. The proposed method is validated in both synthetic and real-world datasets.

**Audience:**

Yes

**Claims And Evidence:**

Yes

**Requested Changes:**

1.	Please provide more details on the environment creation process - how are optimal number of clusters and hyperparameters selected?
2.	Please elaborate more on the time complexity and scalability compared to alternatives like NOTEARS.


**Strengths And Weaknesses:**

Strengths:
1.	The idea of using invariance across environments to learn causal structure is intuitive and reasonable. Environments provide a natural way to identify spurious correlations.
2.	Extending the method to a self-supervised setting is useful, since labeled data is not always available. The two-stage approach seems effective.
3.	This paper is very well written and easy to follow.

Weaknesses:
1.	The evaluation is limited to a few well-known datasets. Testing on a wider variety of datasets would better demonstrate generalizability.
2.	The environment construction process relies on clustering, which may not work well for high-dimensional or complex data.
3.	The clustering for environment construction could be inefficient for large datasets. Analysis of scalability limitations would help understand applicability.

---

> ### Author Response · Authors · 2023-08-23
> **Reply to Reviewer kEJ3 (Part-1)**
>
> Thanks for all your valuable feedback that has helped us to improve our manuscript. Please see the detailed responses below and let us know if you have further questions or comments.
>
> **Weakness 1: Limited Dataset Evaluation**:
>
> We acknowledge that our evaluations were conducted on a few well-known datasets. However, we intentionally and carefully selected these datasets as they have been extensively used as benchmark datasets in the related works [1,2,3,4,5], allowing for direct comparisons of the effectiveness of our approach. Additionally, our evaluation includes both real and synthetic datasets, providing a comprehensive view of the method's applicability. We agree that this is a valuable direction for future work and plan to expand our evaluation in subsequent research.
>
> **Weakness 2: Scalability and Complexity of Environment Construction for High-dimensional Data**:
>
> Thanks for your suggestion. We recognize this challenge and have addressed it in the limitations section.
>
> On the one hand, studying causal relationships for a limited number of variables is more common in real-world scenarios. This is aligned with the trend in various fields where understanding specific relationships is more pertinent than a broader, high-dimensional analysis. Our work acknowledges this trend, and we've designed our methodology consistent with this more focused approach to causal discovery.
>
> 1. In Drug Discovery: Understanding the causal relationship between a specific target protein and disease requires focusing on a limited set of relevant biochemical interactions rather than analyzing hundreds of unrelated variables.
> 2. In Economics: When analyzing the causal impact of interest rates on inflation, economists often consider a well-defined set of macroeconomic variables, rather than all available financial data.
>
> On the other hand, multiple advanced algorithms have addressed the challenge of clustering high-dimensional data. Techniques such as subspace clustering [9], density-based clustering such as DBSCAN [10], and projection-based clustering with techniques like t-SNE and PCA have been developed to tackle this issue.
>
> **Weekness 3: Clustering Scalability**:  In practice, we recognize that clustering for environment construction may present challenges with exceptionally large datasets.
>
> We've designed our approach to typically require only a small number of clusters (ranging from 3 to 5), making the clustering process more manageable. Moreover, the datasets we utilize represent what is commonly used in the community, and our method has been shown to work robustly on them.
>
> Also, clustering can be highly scalable with algorithmic modifications for large datasets. Various works [6, 7, 8] propose scalable and efficient clustering techniques. Such advances can be leveraged in our approach, further extending its applicability to larger datasets.
>
> Furthermore, compared to the iterative process involved in model learning, which may demand extensive computation, the time complexity of our clustering process remains relatively lower, especially with a limited number of clusters.
>
>
> **1. Hyperparameters Tuning for Environment Creation**:
>
> The detailed description of the environment creation process is elaborated in Section 3.2 of the main manuscript, with additional supplementary materials in Section B for comprehensive understanding. In our approach, the optimal number of clusters (K) was determined through an automatic evaluation process.
>
> We employed standard clustering evaluation techniques such as the Elbow and Silhouette Method. These methods enabled us to assess various cluster numbers systematically, identifying the point at which adding more clusters led to diminishing returns (the "elbow") and evaluating how well each object lies within its cluster (the silhouette score).
>
> More specifically, for the Elbow method, we plot the sum of squared distances from each point to its assigned center (known as inertia or within-cluster sum of squares) for a range of values of K (e.g., K from 1 to 10) and automatically detect the "elbow" point on the plot, where the reduction in inertia starts to slow down. For the silhouette method, we calculate its silhouette score, which measures how similar a point is to its cluster compared to other clusters for each data point, and compute the average silhouette score for different values of K and plot them. An algorithm is then applied to look for an optimal value K that maximizes the average silhouette score across all data points.
>
> This process allowed us to arrive at an optimal K value, and our extensive experimentation and analysis demonstrated that our approach remains robust across different numbers of environments. Our empirical evaluations provide further evidence of the effectiveness and stability of our selection method.

---

> > ### Author Response · Authors · 2023-08-23
> > **Reply to Reviewer kEJ3 (Part-2)**
> >
> > **2. Time Comlexity and Scalability Comparision**:
> >
> > With respect to runtime and scalability, Our runtime is 3 to 5 times greater than that of the NOTEAR [1] because of the clustering. Our framework's computational complexity is indeed analogous to that of NOTEARS. Specifically, the complexity for NOTEARS-MLP is given by $O(O(nd^2 m+d^2 m+d^3)$ flops per iteration of L-BFGS-B where n is the number of data sample, d is the number of nodes and m is the number of edges.
> >
> > In our approach, since we divide the dataset into k environments, each requiring separate convergence, the runtime becomes  $O(k (nd^2 m+d^2 m+d^3))$. However, as k (the number of environments) is usually a small constant, typically ranging from 3 to 5, the overall time complexity remains in the same order as in $O(nd^2 m+d^2 m+d^3)$.
> >
> > Some of the quantitative results we have recorded are summarized in the table below.  The time is calculated on Apple M1 Pro chip with 16G memory.
> >
> >
> > | Experiment | NOTEAR time (s) | ISL (ours) | Self-supervised ISL (ours) |
> > |------------|-----------------|------------|----------------------------|
> > | x2s1       | 32.6            | 100.5      | 280.5                      |
> > | x2s2       | 48.1            | 139.7      | 400.2                      |
> > | x3s2       | 62.9            | 188.5      | 520.8                      |
> > | x4s5       | 139.0           | 420.1      | 1220.3                     |
> > | x10s10     | 414.5           | 1300.4     | 3600.4                     |
> >
> >
> >
> > **Reference**
> >
> > [1] Xun Zheng, Chen Dan, Bryon Aragam, Pradeep Ravikumar, and Eric Xing. Learning sparse nonparametric dags. In International Conference on Artificial Intelligence and Statistics, pp. 3414–3425. PMLR, 2020.
> >
> > [2] Xun Zheng, Bryon Aragam, Pradeep K Ravikumar, and Eric P Xing. Dags with no tears: Continuous optimization for structure learning. Advances in Neural Information Processing Systems, 31, 2018.
> >
> > [3] Dennis Wei, Tian Gao, and Yue Yu. Dags with no fears: A closer look at continuous optimization for learning bayesian networks. Advances in Neural Information Processing Systems, 33:3895–3906, 2020
> >
> > [4] Yue He, Peng Cui, Zheyan Shen, Renzhe Xu, Furui Liu, and Yong Jiang. Daring: Differentiable causal discovery with residual independence. In Proceedings of the 27th ACM SIGKDD Conference on Knowledge Discovery & Data Mining, pp. 596–605, 2021.
> >
> > [5] Yue Yu, Jie Chen, Tian Gao, and Mo Yu. Dag-gnn: Dag structure learning with graph neural networks. In International Conference on Machine Learning, pp. 7154–7163. PMLR, 2019.
> >
> > [6] Bahmani, Bahman, et al. "Scalable k-means++." arXiv preprint arXiv:1203.6402 (2012).
> >
> > [7] Zhang, Tian, Raghu Ramakrishnan, and Miron Livny. "BIRCH: an efficient data clustering method for very large databases." ACM sigmod record 25.2 (1996): 103-114
> >
> > [8] Bādoiu, Mihai, Sariel Har-Peled, and Piotr Indyk. "Approximate clustering via core-sets." Proceedings of the thiry-fourth annual ACM symposium on Theory of computing. 2002
> >
> > [9] Agrawal, R.; Gehrke, J.; Gunopulos, D.; Raghavan, P. (2005). "Automatic Subspace Clustering of High Dimensional Data". Data Mining and Knowledge Discovery.
> >
> > [10] Ester, Martin, et al. "A density-based algorithm for discovering clusters in large spatial databases with noise." kdd. Vol. 96. No. 34. 1996

---

### Author Response · Authors · 2023-08-26
**Summary of the Major Changes in the Updated Manuscript**

We would like to thank our reviewers, who put considerable time and thought into helping improve our paper.

We have updated our manuscript. Please find below our summary of the major changes. We have incorporated these major changes into our revised manuscript.

**Summary of major changes**:
1. [nUZd, uRxt, kEJ3] We have detailed how we selected the number of environments we used in the experiment (Appendix Section D).  We specified the number of environments in Section 4.1.2.
2. [uRxt, nUZd] A comprehensive time complexity analysis, along with empirical program runtime metrics, have been incorporated into Appendix Section D.
3. [nUZd, kEJ3] We have revised the methodology section in our manuscript by 1. adding the overview of the methodology section, 2. clarifying the model structure and correcting the typos, and 3. explaining the training details of $h()$
4. [nUZd, kEJ3] Typographical errors in the representation of $\theta^Y_1$ have been rectified. Additionally, Section 3 now provides more details of the neural architecture employed in the model.
5. [nUZd, kEJ3] We have included the discussion of related works on invariant learning and environment-aware partition in Section 2.
6. [nUZd] We have explained why we focused on tabular data in the Appendix Section E

---

### Decision · Action_Editors · 2023-09-28

**Recommendation:** Accept as is

**Comment:**

This paper proposes a framework for discovering causal structures indicating causal relationship by exploiting invariance across environments. This is done by a fairly ingenious pipeline: first use clustering to build sufficiently different environments from a common dataset. Then run causal structure learning on each, producing structures that can be aggregated. In addition, these structures are tied by requiring consistency across predictions when using them as part of model training.

The authors provide convincing results on a set of synthetic and real data experiments, comparing with a variety of causal structure learning methods.

All of the reviewers appreciated the work and saw evidence of the claims. They asked for a number of clarifications and improvements to the writing. The authors provided these and there is consensus for acceptance.

**Audience:**

Yes, problems in causal structure learning are highly relevant to the TMLR audience.

**Claims And Evidence:**

Yes, clear evidence is provided.